# Combining 3D single molecule localization strategies for reproducible bioimaging

Clément Cabriel [1], Nicolas Bourg[1], Pierre Jouchet[1], Guillaume Dupuis[2], Christophe Leterrier [3], Aurélie Baron[4], Marie-Ange Badet-Denisot[4], Boris Vauzeilles[4,5], Emmanuel Fort[6] & Sandrine Lévêque-Fort [1]

Here, we present a 3D localization-based super-resolution technique providing a slowly varying localization precision over a 1 µm range with precisions down to 15 nm. The axial localization is performed through a combination of point spread function (PSF) shaping and supercritical angle fluorescence (SAF), which yields absolute axial information. Using a dual-view scheme, the axial detection is decoupled from the lateral detection and optimized independently to provide a weakly anisotropic 3D resolution over the imaging range. This method can be readily implemented on most homemade PSF shaping setups and provides drift-free, tilt-insensitive and achromatic results. Its insensitivity to these unavoidable experimental biases is especially adapted for multicolor 3D super-resolution microscopy, as we demonstrate by imaging cell cytoskeleton, living bacteria membranes and axon periodic submembrane scaffolds. We further illustrate the interest of the technique for biological multicolor imaging over a several-µm range by direct merging of multiple acquisitions at different depths.

[1] Institut des Sciences Moléculaires d'Orsay, CNRS, Univ. Paris-Sud, Université Paris-Saclay, bâtiment 520, rue André Rivière, 91405 Orsay Cedex, France. [2] Centre de Photonique BioMédicale, Univ. Paris-Sud, Université Paris-Saclay, CNRS, Fédération LUMAT, bâtiment 520, rue André Rivière, 91405 Orsay Cedex, France. [3] Aix-Marseille Université, CNRS, INP, NeuroCyto, 13284 Marseille, France. [4] Centre de Recherche de Gif, Institut de Chimie des Substances Naturelles du CNRS, 91190 Gif-sur-Yvette, France. [5] Laboratoire de Synthèse de Biomolécules, Institut de Chimie Moléculaire et des Matériaux d'Orsay, Univ. Paris-Sud, Université Paris-Saclay, CNRS, 91405 Orsay, France. [6] Institut Langevin, ESPCI Paris, PSL University, CNRS, 1 rue Jussieu, 75005 Paris, France. Correspondence and requests for materials should be addressed to C.C. (email: clement.cabriel@u-psud.fr) or to S.L.-F. (email: sandrine.leveque-fort@u-psud.fr)

D espite recent advances in localization-based super-resolution techniques, nanoscale 3D fluorescence imaging of biological samples remains a major challenge, mostly because of its lack of versatility. While photoactivated localization microscopy (PALM) and (direct) stochastic optical reconstruction microscopy ((d)STORM) can easily provide a lateral localization precision (i.e., the standard deviation of the position estimates) down to 5–10 nm[1–4], a great deal of effort is being made to develop quantitative and reproducible 3D super-localization methods. The most widely used 3D Single Molecule Localization Microscopy (SMLM) technique is astigmatic imaging, which relies on the use of a cylindrical lens to apply an astigmatic aberration in the detection path to encode the axial information in the shape of the spots, achieving an axial localization precision (standard deviation) down to 20–25 nm[5]—though the precision sharply varies with the axial position: 300 nm away from the focus, the precision is typically around 60 nm (see Supplementary Fig. 1a). Other Point Spread Function (PSF) shaping methods are also available[6–8], but their implementations are not as inexpensive and straightforward. Still, all PSF shaping methods including astigmatic imaging suffer from several bias sources such as axial drifts, chromatic aberrations, field-varying geometrical aberrations, and sample tilts. These sources of biases often degrade the resolution or hinder colocalization and experiment reproducibility. Axial measurements can also be performed thanks to intensity-based techniques like Supercritical Angle Fluorescence (SAF)[9–14], which relies on the detection of the near-field emission of fluorophores coupled into propagative waves at the sample/glass coverslip interface due to the index mismatch. Combined with SMLM, this technique, called Direct Optical Nanoscopy with Axially Localized Detection (DONALD) or Supercritical Angle Localization Microscopy (SALM), yields absolute axial positions (i.e., independent of the focus position) in the first 500 nm beyond the coverslip with a precision down to 15 nm[15,16]. The principle relies on the comparison between the SAF and the Undercritical Angle Fluorescence (UAF) components to extract the absolute axial position.

By combining complementary SAF and astigmatism axial information sources, we achieve a slowly varying localization precision over the capture range. Besides, as the SAF detection is insensitive to most axial detection biases inherent in PSF shaping, it provides an absolute reference used to correct the biases of the astigmatic detection. This method, which we call Dual-view Astigmatic Imaging with SAF Yield (DAISY), thus enables reliable and reproducible 3D super-localization imaging of biological samples. It is especially suited for multicolor studies and achieves precisions down to 15 nm.

## Results

**Principle of DAISY and experimental setup**. Starting from the efficient and straightforward astigmatic imaging, we propose to push back its previously mentioned limits; thanks to a novel approach based on a dual-view setup (Fig. 1a) that combines two features. First, it decouples the lateral and axial detections to optimize the 3D localization precision, and second, it uses two different sources of axial information: a strong astigmatism-based PSF measurement is merged with a complementary SAF information that provides an absolute reference. This reference is crucial to render the axial detection insensitive to axial drifts and sample tilts, as well as chromatic aberrations: unlike most other techniques that use fiducial markers[17] or structure correlation[5] to provide these corrections, here, we intend to use the fluorophores themselves as absolute and bias-insensitive references. Besides, by applying a large astigmatic aberration on one fluorescence path only, this technique optimizes the axial precision for the collected photon number (Supplementary Fig. 1b) and maintains a slowly varying localization precision over the imaging depth (Supplementary Fig. 1a). Unlike most PSF shaping implementations found in the literature, which use moderate aberrations[5,18,19] to preserve the lateral resolution, the dual path detection allows one to fully benefit from the astigmatism capabilities. Indeed, as the lateral detection is mostly provided by the aberration-free path, the strong PSF shaping does not compromise the lateral detection. In order to merge the axial and lateral information sources, each is assigned a relative weight according to its localization precision (see Fig. 1b and Methods section). Such a setup exhibits a major improvement in terms of both axial precision and precision curve flatness despite only half of the photons being used for the axial localization far from the coverslip compared with a standard single-view PSF measurement microscope. As a result, DAISY exhibits a weakly anisotropic resolution over the whole capture range.

**DAISY localization precision measurement**. We first performed the calibration of the astigmatism-based axial detection using 15 µm diameter latex microspheres coated with Alexa Fluor (AF) 647 as described in ref. [20] in order to account for the influence of the optical aberrations on the PSFs and thus eliminate this axial bias source (see Methods section). Then, to evaluate the localization precision of DAISY, we imaged dark red 40-nm diameter fluorescent beads located at various randomly distributed heights with a weak 637 nm excitation so that their emission level matched to that of AF647 in typical dSTORM conditions, i.e., 2750 UAF photons and 2750–5100 EPI photons (depending on the depth) per bead per frame on average (Fig. 1c). As it takes advantage of the good performance of the SAF detection near the coverslip, DAISY exhibits a resolution that slowly varies with depth: the lateral and axial precisions reach values as low as 8 nm and 12 nm, respectively (standard deviations), and they both remain better than 20 nm in the first 600 nm. Such precision is sufficient to resolve the hollowness of immunolabeled microtubules, as displayed in Supplementary Fig. 2. This feature is rather uncommon with astigmatic imaging implementations, which typically provide at best 20–25 nm axial precision[5] and only in a limited axial range of ~300 nm according to Cramér-Rao Lower Bound (CRLB) calculations (Supplementary Figs. 1a and 3)—only the dual-objective implementation achieves better precisions, at the cost of a much increased complexity[21]. It is worth noticing that the experimental precisions are slightly worse than the CRLB, which represent a theoretical ideal. This discrepancy is most likely due to optical aberrations, which are not taken into account by the CRLB, and to the use of centroid detection (see Methods section), which is not expected to reach the lower limit.

**Insensitivity to axial detection biases**. Our technique thus provides precise 3D super resolution images (Fig. 1d, e); still, at this precision level, any experimental uncertainty or bias can have devastating effects on the quality of the obtained data. The first source of error that has to be dealt with is the drifts that typically come from a poor mechanical stability of the stage or from thermal drifts. Lateral drifts are well known and can often be easily corrected directly from the localized data using cross-correlation algorithms[22]. However, accounting for the axial drifts can be much more demanding since 3D cross-correlation algorithms require long calculation times unless they sacrifice precision. Tracking fiducial markers is also possible, but since it requires a specific sample preparation and is sensitive to photobleaching (unless a dedicated detection channel at a different wavelength is used[17]), it is not very practical. It is worth noticing

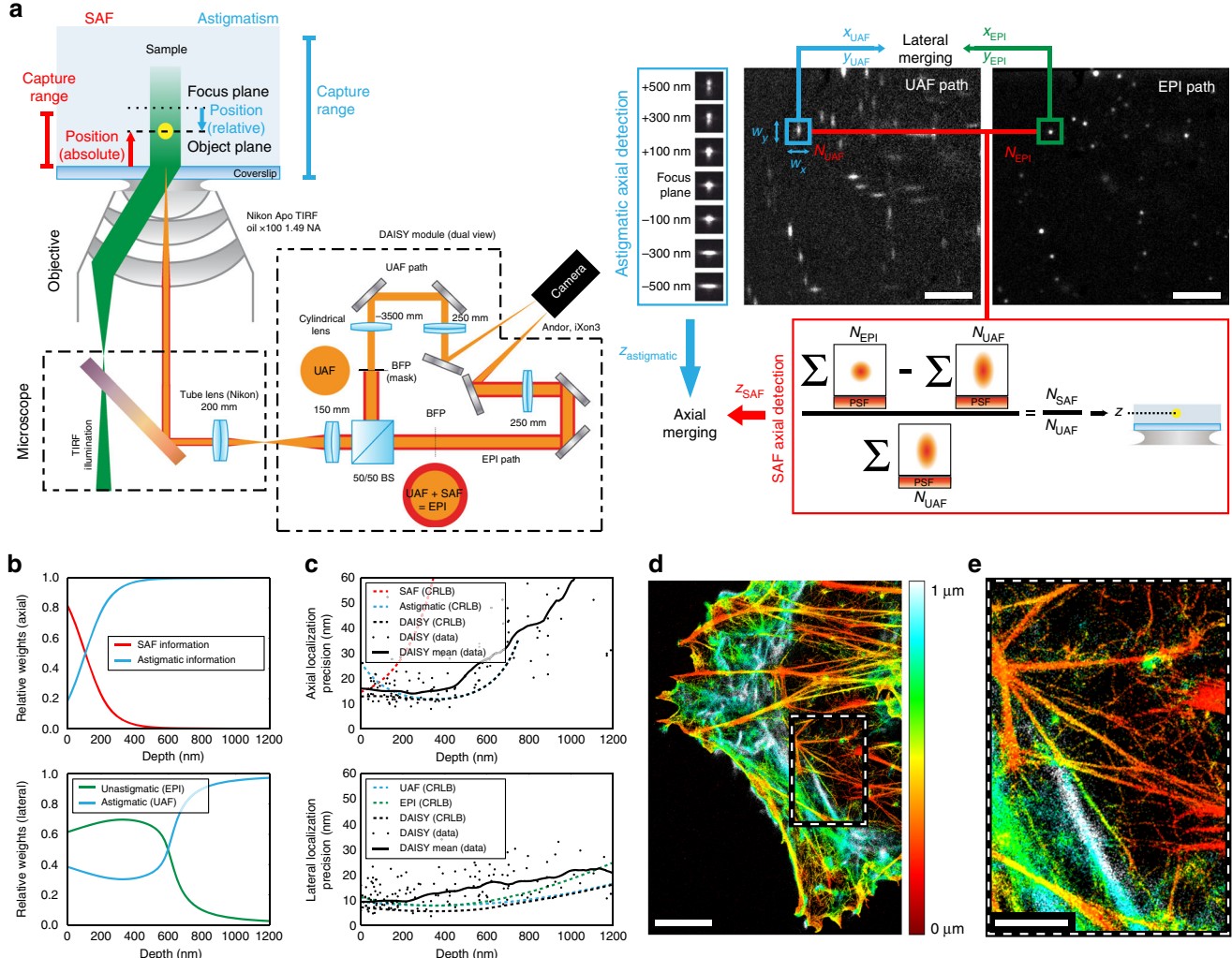

**Fig. 1** Description of the principle of DAISY and characterization of the precision. **a** Schematic of the setup. The DAISY module is placed between the microscope and the camera. After the beam splitter cube (BS), the Undercritical Angle Fluorescence (UAF) path contains a cylindrical lens, as well as a physical mask in a relay plane of the back focal plane of the objective to block the SAF photons. These two elements are not present in the epifluorescence (EPI) detection path, which comprises both the UAF and SAF components. The images are formed on the two halves of the same camera. UAF and EPI frames recorded by the camera on a given field (COS-7 cells, α-tubulin immunolabeling, Alexa Fluor 647) are also displayed (top right corner). For each PSF, the x and y widths are measured to obtain the astigmatic axial information, and the numbers of UAF and EPI photons are used to retrieve the SAF axial information. Finally, the axial astigmatic and SAF positions are merged together. Similarly, lateral positions are obtained by merging the lateral positions from the UAF and EPI paths. **b** Relative weights of the SAF and astigmatic axial detections (top) and of the UAF and EPI lateral positions (bottom) used to merge the positions in DAISY (see Methods section, Position merging section for the exact formulas). **c** Axial (top) and lateral (bottom) precisions of DAISY. The experimental data was taken on dark red 40-nm fluorescent beads distributed at various depths, each emitting a number of photons similar to Alexa Fluor 647. Five-hundred frames were acquired and the precisions were evaluated from the dispersion of the results for each bead. The CRLB contributions of each detection modality are also displayed, as well as the CRLB of DAISY for typical experimental conditions. **d** 3D (color-coded depth) DAISY image of actin (COS-7 cell, AF647-phalloidin labeling). **e** Zoom on the boxed region displayed in **d**. Scale bars: 5 μm (**a**) and (**d**), 2 μm (**e**)

that most commercially available locking systems typically stabilize the focus position at ±30 nm at best (Supplementary Fig. 4), which is hardly sufficient for high resolution imaging. As positions are measured relative to the focus plane with PSF shape measurement methods, axial drifts induce large losses of resolution. On the contrary, SAF detection yields absolute results; thus it is not sensitive to drifts. We use this feature to provide a reliable drift correction algorithm: for each localization, the axial position detected with the SAF and the astigmatism modalities are cross-correlated, which allows us to monitor the focus drift and to consequently correct the astigmatism results with an accuracy typically below 6 nm (see Methods section). To highlight the importance of this correction, we plotted the x–z and y–z profiles of a microtubule labeled with AF647 as a function of time with

both an astigmatism-based detection and DAISY (Fig. 2a–c): unlike the DAISY profiles, the astigmatism profiles exhibit a clear temporal shift, which results in a dramatic apparent broadening of the filament.

In the framework of quantitative biological studies, the axial detection can furthermore be hampered by the axial chromatic aberration due to dispersion by the lenses, including the objective lens. If uncorrected, such a chromatic shift induces a bias in the results of multicolor sequential acquisitions, thus hindering colocalization. However, as DAISY provides absolute axial information, thanks to the SAF measurement, and it is not sensitive to this chromatic aberration. We performed a two-color sequential acquisition on microtubules labeled with AF647 and AF555 (Fig. 2d–f). It illustrates the chromatic dependence

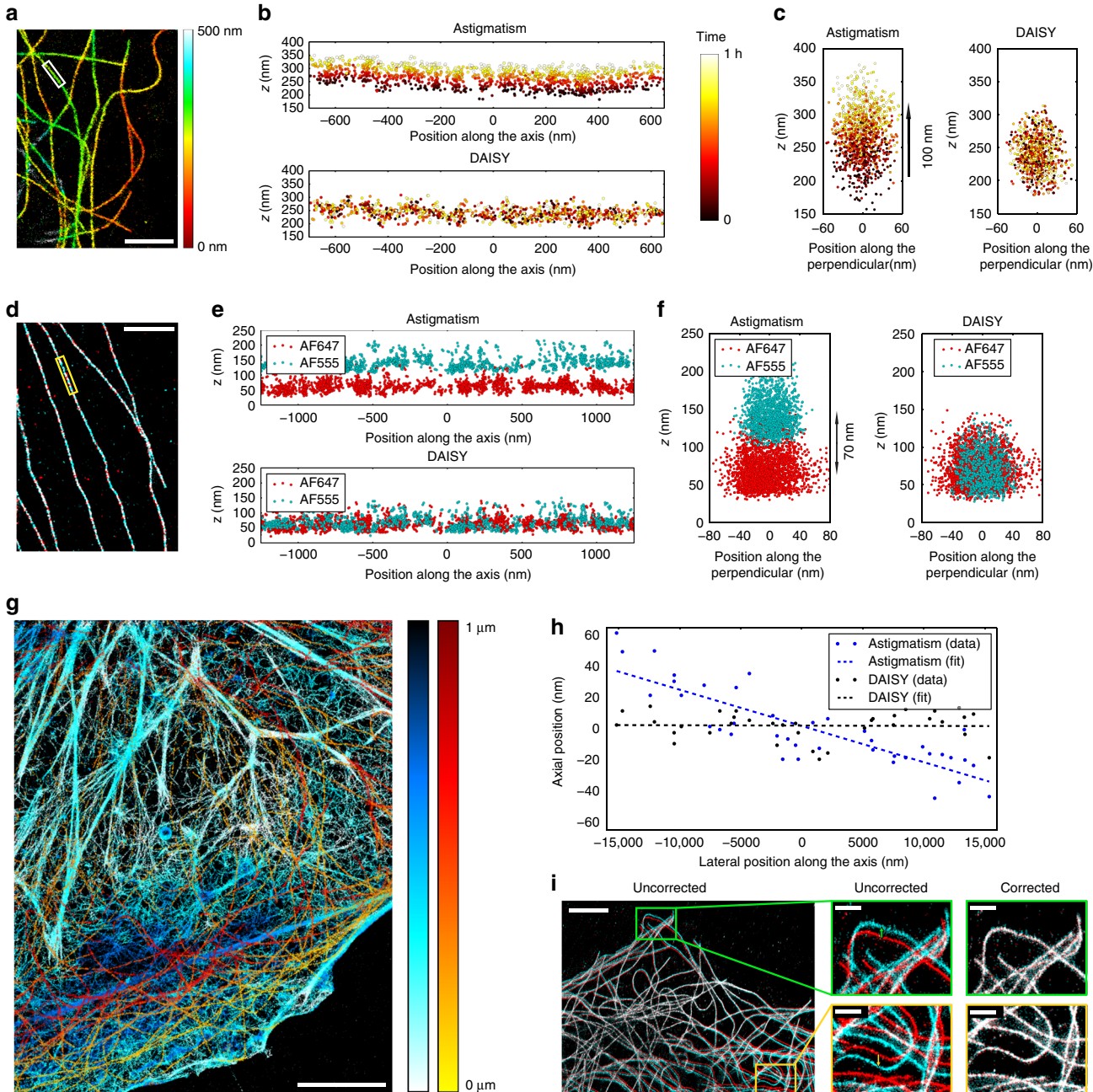

**Fig. 2** Characterization of the performance of DAISY. **a–c** Illustration of the effect of axial drifts. **a** Depth map of microtubules (COS-7 cells, α-tubulin labeled with AF647). The x–z (**b**) and y–z (**c**) profiles of the boxed microtubule are plotted for both standard astigmatic imaging and DAISY. The time is color-coded over 1 h to highlight the effect of the temporal drift. **d–f** Effect of the chromatic aberration. **d** 2D localization image of microtubules (COS-7 cells, α-tubulin labeled with AF555 and β-tubulin labeled with AF647) sequentially imaged in two different colors (red: AF647, cyan: AF555). The x–z (**e**) and y–z (**f**) profiles of the boxed microtubule are plotted for both standard astigmatic imaging and DAISY. **g** Dual-color depth map of actin (cyan-blue) and tubulin (yellow-red) in COS-7 cells (actin labeled with AF647-phalloidin and α-tubulin labeled with a 560-nm excitable DNA-PAINT imager). **h** Influence of the sample tilt on the axial detection. The same field of 20-nm dark red fluorescent beads deposited on a coverslip was imaged with both standard astigmatic imaging and DAISY and the results were averaged over 500 frames to suppress the influence of the localization precision. The detected depth profile is plotted along the tilt axis. **i** Illustration of the image lateral deformation induced by the astigmatism. For the same acquisition (COS-7 cells, α-tubulin labeled with AF647), 2D images were reconstructed from the lateral positions measured on both the astigmatic UAF (in cyan) and the unastigmatic EPI (in red) detection paths of our setup, before the deformation correction (left) and after (right). The whole field and zooms on the boxed regions are both displayed. Scale bars: 2 μm (**a**) and (**d**), 5 μm (**g**) and (**i**) left, 1 μm (**i**) right insets

inherent in standard PSF shaping detection (which exhibits chromatic shifts as large as 70 nm) and the insensitivity of DAISY to this effect (residual chromatic shift inferior to 5 nm). Because of the chromatic shift, the uncorrected astigmatism results appear somewhat inconsistent, whereas the colocalization is much more obvious with DAISY. Consequently, unbiased dual-color 3D images of biological samples can be obtained thanks to sequential acquisitions: we illustrate this on a sample with the actin and the tubulin labeled with AF647 and a 560-nm-excitable DNA-PAINT fluorophore, respectively (Fig. 2g).

It is well known that axial biases in PSF shaping measurements can further stem from tilts of the stage or sample holder, as well as from field-dependent geometrical optical aberrations. These issues were thoroughly studied by Diezmann et al., who reported discrepancies higher than 100 nm over one field of view[23]. Although assessing tilts on biological samples is difficult with PSF measurement methods, DAISY makes this measurement straightforward since the absolute reference provided by the SAF detection can be used to measure the values of the astigmatic axial positions detected for molecules at the coverslip as a function of their lateral positions and then correct the tilt. We performed DAISY acquisitions on 20-nm diameter fluorescent beads at the coverslip and displayed the z values obtained with both an astigmatism-based detection and DAISY. While the former exhibits a clear tilt ranging from −30 to +30 nm over a 30 μm wide field, the latter is insensitive to the tilt, with less than 2 nm axial discrepancy between the two sides of the field (Fig. 2h).

Aside from tilt effects, field-dependent aberrations also induce PSF shape deformations, leading to axial biases. Although we do not actually perform corrections, DAISY is less sensitive to that effect compared with standard astigmatism imaging: on the one hand, the SAF detection relies on intensity measurement, and on the other hand, as DAISY uses a high astigmatism, i.e., strongly aberrated PSFs, it exhibits little sensitivity to remaining field aberrations. To illustrate this phenomenon, we compared tilt-corrected axial positions obtained with 20-nm diameter fluorescent beads deposited on a coverslip between a standard weaker astigmatic detection (350 nm between the two focal lines, close to the values commonly found in the literature) and DAISY. We got rid of the dispersion due to the localization precision by averaging the results over time for each bead and we plotted the corresponding detected depth histograms over one 25-μm wide field of view (Supplementary Fig. 5). The widths of the distributions evidence a much lower impact on the DAISY detection (standard deviation equal to 21 nm) than on the standard astigmatic detection (standard deviation equal to 45 nm). In other words, the strong astigmatism is less sensitive to aberrations than a conventional astigmatism, and the biases are even further mitigated by the coupling with the SAF detection, which relies on photon counting, and is thus weakly sensitive to PSF shapes.

To illustrate the accuracy of the axial correction of the astigmatism data using the SAF measurement, we performed measurements on 40-nm fluorescent beads, both at the coverslip and distributed in the volume (Supplementary Fig. 6). In both cases, the axial correction algorithm seems very accurate (1 nm average discrepancy at the coverslip, and 3 nm in the volume, which is well below the localization precision). As the dispersion of the values increases for beads in the volume, this can be attributed to either the decay of the SAF signal in the volume, which causes the SAF localization precision to become non-negligible, or the influence of the previously mentioned field-dependent aberrations, which induce biases in the astigmatic positions according to the position in the field. This effect is present in conventional single-view PSF shaping imaging too, but it is difficult to detect unless a specifically designed calibration sample is used. The dispersion due to field-dependent aberrations could be mitigated by using a spatially resolved PSF calibration, as in ref. [23].

Lastly, the optical aberrations applied in PSF shaping-based setups not only deform the PSFs, but they may also distort the field itself laterally. For instance, when astigmatism is used, the system has two different focal lengths in x and y, which implies that the magnification is different in x and y. While this effect is of the order of a few percent, it definitely biases the results whenever it is necessary to measure lateral distances precisely unless this magnification discrepancy is duly calibrated. With DAISY, evaluating this image distortion is straightforward—thanks to the non-astigmatic detection path: a cross-correlation performed between the astigmatic (UAF) and the unaberrated (EPI) 2D SMLM images gives the optimal affine transformation to be applied to the astigmatic image—this combination of translation, rotation, and magnification directly provides the magnification difference between the x and y axes, which accounts for 3.5% approximately in our case (Fig. 2i). By applying the optimal affine transformation, the deformation is then corrected: the final lateral discrepancy between the two images was found to be below 6 nm over the whole 25 nm-wide field in Fig. 2i (see Supplementary Fig. 7 for a more detailed measurement of the registration error). It should be noticed, however, that a solution to avoid such a deformation would be to place the cylindrical lens in the Fourier plane, although most reported PSF shaping setups do not use this optical configuration. Also, more complex PSF shapes might induce complex field distortions—potentially making the correction more difficult.

**Multicolor 3D super-resolution imaging of biological samples.** To evidence the performance of DAISY for unbiased, reproducible, and quantitative experiments, we used it to image biological samples. We illustrate the performances in terms of resolution by performing acquisitions on living E. coli bacteria adhered to a coverslip. The envelope of bacteria was labeled with both AF647 and AF555 using a click chemistry process (see Methods section)[24,25]. Since the lipopolysaccharide (LPS) layer is thin in Gram-negative bacteria, this is a good sample to observe the influence of the localization precision. We present in Fig. 3a, b 2D and 3D images of a region of interest and in Fig. 3c an x–z slice along the line displayed in Fig. 3a. The measured diameter of the bacterium is around 1 μm but still it does not exhibit a strong loss of resolution at its edges. To evidence this, we also plotted the lateral and axial histograms in the boxed regions (Fig. 3c). The axial standard deviations were found to be, respectively, around 30 nm and 45 nm at the bottom and at the top of the cell, while lateral standard deviations were around 27 nm in both colors. Taking into account the size of the LPS layer (<10 nm), of the label—i.e., the DBCO-sulfo-biotin and streptavidin-AF construction—(10 nm) and the effect of the curvature of the bacterium over the width of the area used for the analysis (10 nm), these values are consistent with the localization precision curves plotted in Fig. 1c. As a comparison, the results obtained on the same sample with uncorrected astigmatism and with DONALD are provided in Supplementary Fig. 8. Like DAISY, DONALD features an absolute detection, unsensitive to both chromatic aberration and axial drift. However, the axial precision of DONALD deteriorates sharply with the depth due to the decay of the SAF signal; thus the top half of the sample (beyond 500 nm) is hardly visible, whereas DAISY clearly permits imaging up to 1 μm. Uncorrected astigmatism has the same capture range as DAISY, but since it lacks the absolute information, it exhibits an axial shift between the two colors, as well as a broadening of the histogram widths due to the axial drift.

We then used DAISY to visualize the periodic submembrane scaffold present along the axon of cultured neurons[26]. We imaged the 3D organization of two proteins within this scaffold: adducin (labeled with AF647) that associates with the periodic actin rings, and β2-spectrin (labeled with AF555) that connect the actin rings (Fig. 3d–f). The lateral resolution allowed us to easily resolve the alternating patterns of adducin rings and β2-spectrin epitopes and their 190 nm periodicity (Fig. 3g)[27]. Thanks to the axial resolution of DAISY, we were also able to resolve the submembrane localization of both proteins across the whole diameter of the axon at 600 nm depth (Fig. 3h).

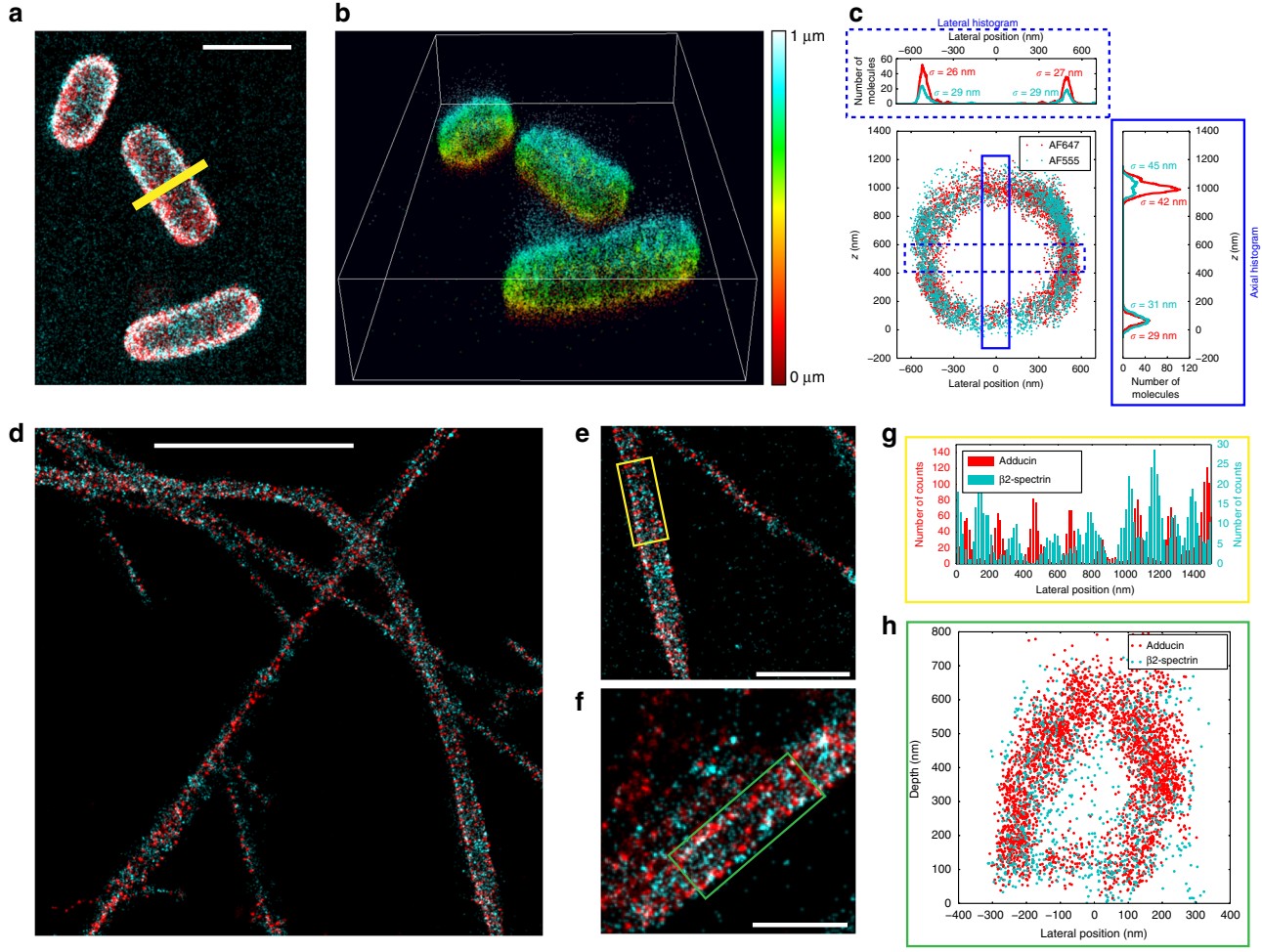

**Fig. 3** DAISY results obtained from biological samples. **a** 2D SMLM image of living *E. coli* bacteria labeled with both AF647 (red) and AF555 (cyan) at the membrane. **b** 3D view of the field displayed in **a**. The depth is color-coded (one single colormap is used for both AF647 and AF555). **c** *x–z* slice along the line displayed in **a** and axial and lateral profiles in the boxed regions. The σ values stand for the standard deviations of the distributions. **d–f** 2D dual-color images of rat hippocampal neurons where the adducin and the *β*2-spectrin were labeled with AF647 and AF555, respectively. **g** Lateral profile along the axis of the yellow box displayed in **e**. **h** *x–z* slice along the green box displayed in **f**. Scale bars: 2 μm (**a**) and (**e**), 5 μm (**d**), 1 μm (**f**)

**Extended depth imaging**. Taking advantage of the features of DAISY for unbiased sequential imaging, we propose an implementation allowing single-color and multicolor imaging at wider depth ranges by stacking the results of multiple acquisitions on the same field at different heights. Although PSF measurement methods also allow this type of acquisitions, DAISY is especially suited in this case, thanks to its previously described intrinsic bias correction features. Since the SAF signal quickly decays with the depth in the first 500 nm above the coverslip, the absolute reference is accessible only in the first stack. Still, as it provides unbiased results, the top of this first stack serves as an absolute reference for the next stack, which is matched to the previous using an axial position cross-correlation algorithm. In other words, the first 1 μm unbiased slice is interlaced with the following one, which contains the positions between 600 nm and 1.6 μm (as described in the schematic in Fig. 4a). The absolute reference is thus transferred from the first slice onto the second, which becomes insensitive to axial detection biases. Similarly, the third slice, containing positions from 1.2 to 2.2 μm is intertwined with the second by position cross-correlation, and thus it also benefits from the absolute reference and the bias insensitivity that it brings. Several slices can be recorded and merged together to obtain an extended depth image—still, this is limited by photobleaching (although this can be mitigated by using (DNA-)

PAINT labeling), as well as aberrations inherent in depth imaging, which cause the axial and lateral precisions to deteriorate away from the coverslip. Moreover, registration errors are likely to add as the number of slices increases, so using fiducial markers might be necessary to merge more slices. We illustrate the method with a single-color acquisition series (COS-7 cells, *α*-tubulin and *β*-tubulin labeled with AF647) in Fig. 4b–d: the stack of the three slices (Fig. 4e) obviously shows information in deep regions (beyond 1 μm) that would not be accessible with a single acquisition. We then imaged a dual-label tubulin-clathrin sample (COS-7 cells, light chain and heavy chain clathrin labeled with AF647, *α*-tubulin and *β*-tubulin labeled with 560-nm-excitable DNA-PAINT imager) in three sequential acquisitions while shifting the focus by 600 nm between each of them to obtain a 3D dual-color 2 μm imaging range set of data (Fig. 4f). Aside from the fact that no axial mismatch between the subsequent acquisitions is observed, the localization precision remains satisfactory after 1.5 μm as it is limited only by the effect of the spherical aberration and sample-induced aberrations. To evidence this, we measured the dispersion of the localizations on two clathrin spheres located close to the ventral membrane (200 nm depth) and the dorsal membrane (1500 nm depth), respectively (Fig. 4g–h, Supplementary Fig. 9). The lateral and axial standard deviations were found to be 16 nm in *xy* and 17 nm in *z* at

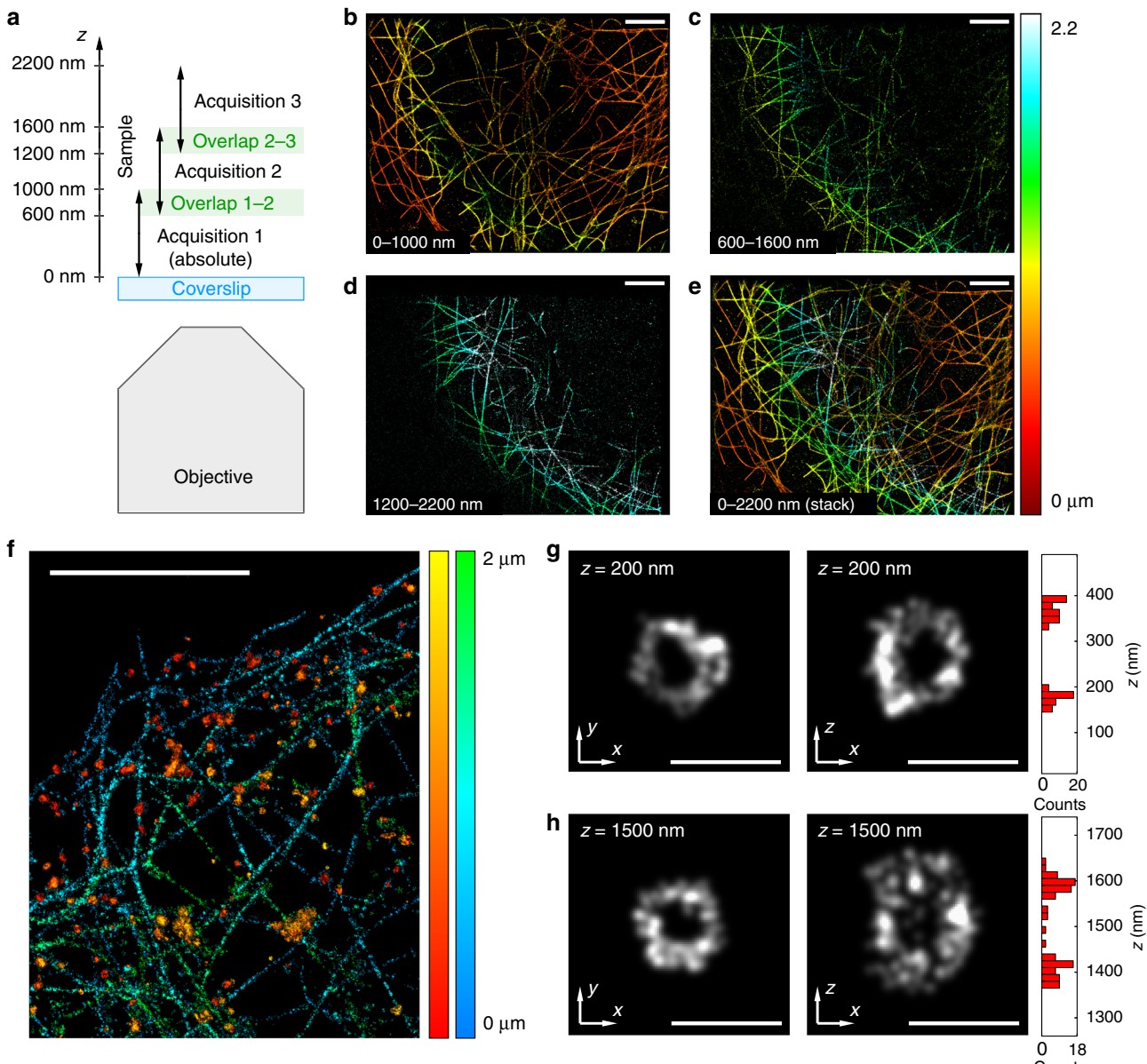

**Fig. 4** Extended depth imaging principle and results. **a** Description of the acquisition protocol: several sequential acquisitions are performed at different focus positions with a sufficient overlap between them to enable the stitching of the different slices (the focus is typically shifted by 600 nm between successive acquisitions, while the capture range is around 1 μm for each acquisition). **b**–**d** 3D images reconstructed from single-color tubulin acquisitions performed at three different focus positions (COS-7 cells, α-tubulin and β-tubulin labeled with AF647). **e** Final 3D image obtained by stitching the three consecutive acquisitions. The total range is around 2.2 μm. **f** 3D extended range dual-color image of clathrin (red-yellow) and tubulin (blue-green) obtained from three sequential acquisitions (for each color) at different heights (COS-7 cells, heavy chain and light chain clathrin labeled with AF647, α-tubulin and β-tubulin labeled with a 560-nm excitable DNA-PAINT imager). **g**, **h** x–y and x–z slices of two clathrin spheres taken from (**f**) at two different depths (200 and 1500 nm). The axial histograms of the x–z images are displayed on the right. Scale bars: 5 μm (**b**–**f**), 250 nm (**g**, **h**)

200 nm depth, and 20 nm in *xy* and 27 nm in *z* at 1500 nm depth—as expected, the axial precision is more affected by the effect of the aberrations in the volume than the lateral precision.

## Discussion

Thanks to the decoupling of the axial and lateral detections and to the combination of two axial SMLM techniques yielding complementary information, we could achieve reliable and unbiased imaging that enables quantitative studies on biological samples. DAISY offers a slowly varying, weakly anisotropic resolution over the whole micron-wide capture range, with a localization precision down to 15 nm. Thanks to both the SAF and the astigmatic

detections, DAISY provides absolute axial results that prove to be insensitive to axial drifts and sample tilts, as well as chromatic aberration. These features make it especially suited for biological samples imaging near the coverslip, which finds applications in the framework of cell adhesion, motility processes, bacteria imaging or neuronal axons and dendrites studies. Moreover, stacking acquisitions performed at different heights also enables reproducible and reliable studies at more important depths, upto a few micrometers. Finally, as the implementation of the dual-view detection scheme we use is straightforward, it would also benefit any PSF measurement method other than astigmatism, such as double-helix PSF[6], self-bending PSF[7], saddle-point PSF[8],

and tetrapod[28], which offer better performances in terms of localization precision and capture range.

## Methods

**Optical setup**. A schematic of the optical setup used is presented in Fig. 1a. We used a Nikon Eclipse Ti inverted microscope with a Nikon Perfect Focus System. The excitation was performed thanks to five different lasers: 637 nm (Obis 637LX, 140 mW, Coherent), 561 nm (Genesis MX 561 STM, 500 mW), 532 nm (Verdi G5, 5 W, Coherent), 488 nm (Genesis MX 488 STM, 500 mW, Coherent), and 405 nm (Obis 405LX, 100 mW, Coherent). The corresponding 390/482/532/640 or 390/482/561/640 multiband filters (LF405/488/532/635-A-000 and LF405/488/561/635-A-000, Semrock) were used. The fluorescence was collected through a Nikon APO TIRF ×100 1.49 NA oil immersion objective lens, sent in the DAISY module and recorded on two halves of a 512 × 512-pixel EMCCD camera (iXon3, Andor). The camera was placed at the focal plane of the module of magnification 1.67 and the optical pixel size was ~100 nm. Finally, the imaging paths were calibrated in intensity to compensate the non-ideality of the 50–50 beam splitter, as well as the reflection on the cylindrical lens surface (this measurement was performed for each fluorescence wavelength). The object focal plane of the EPI path was typically at the coverslip ($z = 0$ nm) and the UAF path had two focal lines, at $z = 0$ nm and $z = 800$ nm for the $y$ and $x$ axes, respectively.

**Sample preparation**. COS-7 cells were grown in DMEM with 10% FBS, 1% L-glutamin, and 1% penicillin/streptomycin (Life Technologies) at 37 °C and 5% $CO_2$ in a cell culture incubator. Several days later, they could be plated at low confluency on cleaned round 25 mm diameter high resolution 1.5H glass coverslips (Marienfield, VWR). After 24 h, the cells were washed three times with PHEM solution (60 mM PIPES, 25 mM HEPES, 5 mM EGTA, and 2 mM Mg acetate adjusted to pH 6.9 with 1 M KOH) and fixed for 12 min in 4% PFA, 0.2% glutaraldehyde and 0.5% Triton; they were then washed 3 times in PBS (Invitrogen, 003000). Upto this fixation step, all chemical reagents were pre-warmed at 37 °C. The cells were post-fixed for 10 min with PBS + 0.1% Triton X-100, reduced twice for 10 min with NaBH₄, and washed in PBS three times before being blocked for 15 min in PBS + 1% BSA.

The labeling step varied according to the required sample: in the case of actin labeling, the cells were incubated for 20 min with 3.3 nM phalloidin-AF647 (Thermo Fisher, A22287) in the dSTORM imaging buffer (Abbelight) before starting the acquisition—without removing the dSTORM buffer containing the phalloidin-AF647. On the contrary, immunolabeling of tubulin and clathrin required more preparation steps.

For AF647 α-tubulin, the cells were incubated for 1 h at 37 °C with 1:300 mouse anti-α-tubulin antibody (Sigma Aldrich, T6199) in PBS + 1% BSA. This was followed by three washing steps in PBS + 1% BSA, incubation for 45 min at 37 °C with 1:300 goat anti-mouse AF647 antibody (Life Technologies, A21237) diluted in PBS 1% BSA and three more washes in PBS.

For AF647 β-tubulin and AF555 α-tubulin, the cells were incubated for 1 h at 37 °C with 1:300 rabbit anti-β-tubulin antibody (Sigma Aldrich, T5293) in PBS + 1% BSA. This was followed by three washing steps in PBS + 1% BSA, incubation for 45 min at 37 °C with 1:300 goat anti-rabbit AF555 antibody (Life Technologies, A21430) diluted in PBS + 1% BSA and three more washes in PBS + 1% BSA. Then they were incubated again for 1 h at 37 °C with 1:300 mouse anti-α-tubulin antibody (Sigma Aldrich, T6199) in PBS + 1% BSA, washed three times, incubated for 45 min at 37 °C with 1:300 goat anti-mouse AF647 antibody (Life Technologies, A21237) in PBS + 1% BSA and washed three more washes in PBS.

For AF647 α-tubulin and β-tubulin, the cells were incubated for 1 h at room temperature with 1:300 mouse β-tubulin antibody (Sigma Aldrich, T5293) in PBS + 1% BSA. This was followed by three washing steps in PBS + 1% BSA, incubation for 1 h at 37 °C with 1:300 mouse α-tubulin antibody (Sigma Aldrich, T6199) diluted in PBS 1% BSA, three more washes in PBS + 1% BSA, incubation for 45 min at 37 °C with 1:300 goat anti-mouse AF647 antibody (Life Technologies, A21237) diluted in PBS 1% BSA and three more washes in PBS.

For AF647 heavy chain and light chain clathrin and DNA-PAINT α-tubulin and β-tubulin, the cells were incubated for 1 h at 37 °C with 1:400 mouse anti-light chain clathrin antibody (Sigma Aldrich, C1985) in PBS + 1% BSA and washed three times with PBS + 1% BSA, incubated again for 1 h at 37 °C with 1:400 mouse anti-heavy chain clathrin antibody (Sigma Aldrich, C1860) in PBS + 1% BSA and washed three times with PBS + 1% BSA. Then, they were incubated for 45 min at 37 °C with 1:400 anti-mouse AF647 antibody (Life Technologies, A21237) in PBS + 1% BSA, washed three times with PBS + 1% BSA, and incubated again for 1 h at room temperature with 1:400 mouse β-tubulin antibody (Sigma Aldrich, T5293) in PBS + 1% BSA. This was followed by three washing steps in PBS + 1% BSA, incubation for 1 h at 37 °C with 1:400 mouse α-tubulin antibody (Sigma Aldrich, T6199) diluted in PBS 1% BSA, three more washes in PBS + 1% BSA, incubation for 2 h at 37 °C with 1:100 anti-mouse-D1 Ultivue secondary antibody diluted in antibody dilution buffer (Ultivue-2 kit, Ultivue) and washed three more washes in PBS.

In any case, after the immunolabeling of tubulin and/or clathrin, a post-fixation step was performed using PBS with 3.6% formaldehyde for 15 min. The cells were washed in PBS three times and then reduced for 10 min with 50 mM NH₄Cl (Sigma Aldrich, 254134), followed by three additional washes in PBS.

To prepare the neuron samples, rat hippocampal neurons from E18 pups were cultured on 18 mm coverslips at a density of 6000 cm⁻² according to previously published protocols[29] and following guidelines established by the European Animal Care and Use Committee (86/609/CEE) and approval of the local ethics committee (agreement D18-055-8). After 16 days in culture, neurons were fixed using 4% PFA in PEM (80 mM Pipes, 5 mM EGTA, and 2 mM MgCl₂, pH 6.8) for 10 min. After rinsing in 0.1 M phosphate buffer (PB), neurons were blocked for 60 min at room temperature in immunocytochemistry buffer (ICC: 0.22% gelatin, 0.1% Triton X-100 in PB). Following this, neurons were incubated with a chicken primary antibody against map2 (abcam, ab5392) mouse primary antibody against β2-spectrin (BD Bioscience, 612563) and a rabbit primary antibody against adducin (abcam, ab51130) diluted in ICC overnight at 4 °C, then after ICC rinses with AF 488, 555, and 647 conjugated secondary antibodies for 1 h at 23 °C.

The *E. coli* K12 (MG1655) cells were grown in 2YT medium (Sigma, Tryptone 16.0 g.L⁻¹, Yeast extract 10.0 g.L⁻¹, NaCl 5.0 g.L⁻¹) at 37 °C under agitation (180 rpm). Overnight cultures were diluted 100 times in fresh medium (final volume 300 μL) containing Kdo-N₃ (1.0 mM). Bacteria were incubated at 37 °C for 9 h under agitation (180 rpm). Then 200 μL of the obtained suspension were washed 3 times with PBS buffer (200 μL, 9700 × g, 1 min, room temperature). The pellet was re-suspended in 200 μL of a solution of DBCO-Sulfo-Biotin (JenaBioscience, CLK-A116) (0.50 mM in PBS buffer) and the suspension was vigorously agitated for 90 min at room temperature. Bacteria were washed 3 times with PBS buffer (200 μL, 9700 × g, 1 min, room temperature). The pellet was re-suspended in a solution of Streptavidin-AF647/Streptavidin-AF555 (20 μg.mL⁻¹ each) (Invitrogen, ThermoFischer Scientific, S21374 and S32355) in PBS containing BSA (1.0 mg.mL⁻¹, 200 μL) and the suspension was agitated at room temperature for 90 min in the dark. Bacteria were then washed 3 times with PBS buffer (200 μL, 9700×g, 1 min, room temperature). The pellet was re-suspended in PBS buffer (400 μL) and stored at 4 °C until analysis.

**Fluorescent beads sample preparation**. Twenty-nanometer fluorescent dark red beads samples (Fig. 2h, Supplementary Fig. 5) were prepared using a $5.10^{-7}$ dilution of the initial solution (F8783, Thermo Fisher). We performed the dilution in PBS + 5% glucose to match the index of the dSTORM imaging buffer, and we waited for 5 min before starting the acquisition so that the beads had time to deposit on the coverslip.

Hundred-nanometer diameter tetraspeck fluorescent beads samples (Supplementary Fig. 4) were prepared by diluting the initial solution (T7279, Thermo Fisher) at $5 \times 10^{-4}$ in PBS + 5% glucose, and we waited for 5 min before starting the acquisition for the beads to deposit on the coverslip.

The samples of 40-nm diameter dark red fluorescent beads deposited on a coverslip (Supplementary Figs. 6a and 7a–c) were obtained by diluting the initial solution (10720, Thermo Fisher) at $5 \times 10^{-7}$ in PBS + 5% glucose, and we waited for 5 min before starting the acquisition for the beads to deposit on the coverslip.

The samples of 40-nm diameter dark red fluorescent beads randomly distributed in the imaging volume (Fig. 1c, Supplementary Fig. 6b) were obtained by taking fixed, unlabeled COS-7 cells and adding 500 μL of beads solution (10720, Thermo Fisher) diluted at $5 \times 10^{-7}$ in PBS during 5 min for beads to deposit before removing the solution and replacing it with PBS + 5% glucose. Beads stuck on the upper side of the membrane were thus located at random heights.

**Image acquisition**. dSTORM and DNA-PAINT imaging on biological samples was performed using an oblique epifluorescence illumination configuration. To induce most of the molecules in a dark state, we used a dSTORM buffer (Abbelight Smart kit). The sample was lit with an irradiance of ~4 kW.cm⁻² until a sufficient density of molecules was obtained—typically below one molecule per 4 μm² (see Supplementary Note 1 for a study of the influence of the molecule density per frame on the localization performance). We then started the data acquisition with a 50-ms (for AF647) or 100-ms (for AF555) exposure time and 150 EMCCD gain. The total number of acquired frames was typically between 15,000 and 30,000 per acquisition.

For sequential dSTORM and DNA-PAINT acquisitions, the dSTORM acquisition was first performed as described above. Then, we removed the dSTORM buffer and added a 0.5 nM dilution of DNA-PAINT imagers in imaging buffer (I1-560, Ultivue-2 kit, Ultivue). To achieve single molecule regime, the sample was lit with an irradiance of ~4 kW.cm⁻² and we then started the data acquisition with a 100-ms exposure time and 150 EMCCD gain. The total number of acquired frames was around 50,000.

Performance measurements on fluorescent beads were done at low illumination powers (0.15 kW.cm⁻² for 20-nm diameter dark red beads and 0.025 kW.cm⁻² for tetraspeck beads and 40-nm diameter dark red beads). The beads were immersed in PBS + 5% glucose and the exposure times and EMCCD gain were 50 ms and 150 ms, respectively. Except for the long-term axial drift tracking experiment, 500 frames were recorded for each performance characterization acquisition.

The acquisition was performed using the Nemo software (Abbelight).

**Localization software**. Each 512 × 512-pixel frame was pre-processed by removing the pixel per pixel temporal median of the previous 10 frames in order to get rid of the slowly varying background without altering the number of photons in the PSFs.

The filtered frames were then split in two parts corresponding to the UAF and EPI paths of the DAISY module, respectively. On the $512 \times 256$-pixel sub-frames, the PSFs were detected using a second order wavelet filtering associated with an intensity threshold (typically 1.0 for the EPI channel, 0.8 for the UAF channel). Each PSF was characterized using a center of mass detection to retrieve the lateral positions $x^{EPI}$, $y^{EPI}$, $x^{UAF}$, and $y^{UAF}$, and a Gaussian fitting to assess the PSF widths $w_x^{UAF}$, $w_y^{UAF}$, $w_x^{EPI}$, and $w_y^{EPI}$. A photon counting was also performed over a $2 \times 2$ μm square area centered on the PSF to determine the number of photons $N^{EPI}$ and $N^{UAF}$. A filtering step based on photon numbers (typically 500 photons minimum for AF647), EPI PSF widths (80 nm $< \sqrt{w_x^{EPI} w_y^{EPI}} < 180$ nm) and EPI PSF anisotropy ($0.67 < w_x^{EPI}/w_y^{EPI} < 1.5$) was then operated to get rid of false positive detections. Furthermore, pairs of localizations closer than 2 μm were discarded to avoid biases due to the signal from neighboring PSFs. Corrections were applied to photon numbers (as mentioned in the Optical setup section) and lateral positions $x^{UAF}$ and $y^{UAF}$ (to compensate the image deformation induced by the astigmatism as illustrated in Fig. 2i and Supplementary Fig. 7). Afterwards, the axial positions were calculated: the values of $z^{SAF}$ were obtained using the theoretical curve provided in ref. [15] whereas those of $z^{astigmatic}$ could be retrieved by fitting $w_x^{UAF} - w_y^{UAF}$ to the calibration curve (see the Astigmatism calibration section) using a least squares calculation. Lateral drifts were then corrected using a temporal cross-correlation algorithm. Furthermore, $z^{astigmatic}$ positions were corrected using the SAF reference (see Astigmatism correction algorithm section).

Finally, the values of $z^{SAF}$ and $z^{astigmatic}$ were merged together, as well as the values of $x^{EPI}$ and $x^{UAF}$, $y^{EPI}$ and $y^{UAF}$ (as described in the Position merging section).

All this processing was performed using a home-written Python code.

**Astigmatism calibration**. Although in the literature, the calibration of axial detection methods is often performed by using fluorescent beads deposited on a coverslip and defocusing the objective, this method is biased since it does not take into account the effect of the spherical aberration, which affects both the position of the focal plane (the so-called focal shift) and the shapes of the PSFs. While the former can be compensated using a calculated correction factor depending on several experimental parameters, there is no simple way to get to correct the latter to our knowledge. Thus, we chose to perform the calibration of the astigmatic detection using a sample of known geometry in the nominal acquisition conditions, i.e., with a fixed focus and dSTORM fluorophores. More specifically, we used a sample of 15 μm microspheres decorated with fluorophores (either AF647 or AF555), as described in ref. [20]. By measuring the position of the center and the radius of the spheres, it is possible to calculate the expected axial position of each molecule from the measurement of its lateral position. Such an acquisition provides the lookup table giving the correspondence between PSF widths and axial positions.

**Astigmatism correction algorithm**. Before combining the two sources of axial information, the astigmatic positions were corrected in order to make them benefit from the SAF absolute detection. This was completed, thanks to a cross-correlation algorithm between the SAF and astigmatic positions measured for each molecule. As the SAF detection is efficient mostly close to the coverslip, we restricted the data to the subset of molecules verifying $z^{SAF} \in [-50$ nm, 300 nm$]$ in order to perform the cross-correlation in the domain where both axial information sources are precise and reliable.

First, we removed the tilt: the $z^{SAF} - z^{astigmatic}$ axial discrepancy was calculated for each molecule from the data verifying $z^{SAF} \in [-50$ nm, 300 nm$]$. The spatially resolved axial discrepancy information was used to calculate the tilt by fitting a plane to the data, which provided the tilt direction and amplitude. The astigmatic positions were corrected in accordance with this result.

Then data was divided in subsets of 1000 frames and distributed in series of 3D images with 100 nm lateral and 15 nm axial pixel sizes, each of them corresponding to a 1000 frame subset. For each subset, the SAF and astigmatism 3D images were cross-correlated allowing only axial displacements to maximize the overlap, which brought the correction to be applied to the astigmatic positions for the subset. Then, the results obtained for all the subsets were pooled and interpolated to generate the axial drift curve. Thanks to this correction, the astigmatic results were made absolute (i.e., referenced to the coverslip) and insensitive to both the chromatic aberration and the axial drift.

It is worth noting that the 1000-frame division corresponds to a 50-s sampling of the axial drift (with 50-ms exposure time). This value seems reasonable given the slow evolution of the drift: it is the result of a compromise between the bandwidth of the correction (a finer sampling allows a better correction of higher drift frequencies) and the robustness of the algorithm (if the amount of data is too low, the algorithm may not adequately converge or provide a wrong value). Shorter slices might be used with higher density samples. Similarly, acquisitions featuring a lower SNR or photon number would require larger pixels or larger slices to compensate the influence of the localization precision worsening. The final accuracy of the correction appears to be typically better than 3 nm (this was obtained by measuring the height of fluorophores deposited at the coverslip outside of cells).

**Position merging**. In DAISY acquisitions, the lateral positions were obtained by combining the two sources of lateral information according to their uncertainties (the CRLB values were used for that purpose). The exact formula follows the normal distribution combination law:

$$x^{DAISY} = \left( \frac{x^{UAF}}{(\sigma_x^{UAF})^2} + \frac{x^{EPI}}{(\sigma_x^{EPI})^2} \right) \Big/ \left( \frac{1}{(\sigma_x^{UAF})^2} + \frac{1}{(\sigma_x^{EPI})^2} \right) \quad (1)$$

$$y^{DAISY} = \left( \frac{y^{UAF}}{(\sigma_y^{UAF})^2} + \frac{y^{EPI}}{(\sigma_y^{EPI})^2} \right) \Big/ \left( \frac{1}{(\sigma_y^{UAF})^2} + \frac{1}{(\sigma_y^{EPI})^2} \right) \quad (2)$$

where $\sigma_i^{UAF}$ and $\sigma_i^{EPI}$ are the localization precisions in the direction $i$ for the UAF and EPI detections, respectively (i.e., the standard deviations of the positions).

Similarly, the two sources of axial information were merged according to their uncertainties:

$$z^{DAISY} = \left( \frac{z^{SAF}}{(\sigma_z^{SAF})^2} + \frac{z^{astigmatic}}{(\sigma_z^{astigmatic})^2} \right) \Big/ \left( \frac{1}{(\sigma_z^{SAF})^2} + \frac{1}{(\sigma_z^{astigmatic})^2} \right) \quad (3)$$

where $\sigma_z^{SAF}$ and $\sigma_z^{astigmatic}$ are the axial localization precisions of the SAF and the astigmatic detections, respectively.

This combination optimizes the final precision, i.e., it provides the best precision attainable from the two sources given their respective uncertainties.

The relative weights used for DAISY are shown in Fig. 1b. It is worth noting that since localization precisions vary with depth, the corresponding weights vary accordingly. Notably, the weight of the SAF detection is more important than that of the axial astigmatic detection at the coverslip, but it quickly dwindles to almost zero after 500 nm. Similarly, the (unastigmatic) EPI detection is more precise in the first depth of field, whereas the (astigmatic) UAF detection dominates after 600 nm, where the EPI PSFs are too defocused to be detected.

**Localization precision measurement**. To obtain the localization precisions displayed in Fig. 1c, we prepared a sample of 40-nm dark red fluorescent beads randomly distributed in the imaging volume (see Fluorescent beads sample preparation section). The results of several 500-frame acquisitions were pooled and for each of them, the lateral drift was corrected. The average axial position was measured for each bead, as well as the standard deviations on the lateral and axial measured positions, which gave the localization precisions. The laser power was adjusted so that the photon numbers emitted by the beads matched those of AF647 (2750 UAF photons per PSF and 2750–5100 EPI photons, depending on the depth of the bead).

Using fluorescent beads seems to be a more reliable method to measure the localization precisions than with biological samples—unlike the fluorescent beads, the use of biological samples requires many assumptions on the size and geometry of the labeled target, the label (which is typically around 10–15 nm in the case of immunolabeling), the fluorophore itself, as well as the motion freedom of the label.

**Cramér-Rao lower bound calculation**. To derive the CRLB for DAISY, we first estimated the lower bounds associated to the astigmatic and the SAF detections separately. To this end, we assumed elliptical Gaussian PSFs for the UAF image and circular Gaussian PSFs for the EPI image. We used a realistic set of parameters corresponding to typical experimental conditions with AF647, i.e., 100 background photons per pixel on each path and a number of photons per PSF equal to 2750 for the UAF path and 2750–5100 for the EPI path (depending on the axial position). The CRLB of the SAF was adapted from[30] and that of the astigmatism was derived from[31]. Finally, the DAISY axial CRLB was obtained from the previous results using Eq. (3). Similarly, the lateral CRLB for the UAF and EPI paths were obtained from[32] and the lateral lower bound of DAISY was calculated from these results using Eqs. (1) and (2). See Supplementary Note 2 for a more exhaustive description of the CRLB calculations. These results were used to plot the curves displayed in Fig. 1c and Supplementary Figs. 1 and 3.

Note that the CRLB values are somewhat optimistic and that they are not necessarily expected to be reached in real experimental conditions because they do not account for optical aberrations, polarization effects on the PSF shape or for the ability of the localization algorithm to actually extract the best possible information.

**Data visualization**. The 3D view in Fig. 3b was obtained using the Nemo software (Abbelight).

A filter based on the local density of molecules associated with a threshold was applied on Fig. 4f–h to remove false positive detections.

**Reporting summary**. Further information on research design is available in the Nature Research Reporting Summary linked to this article.

## Data availability

Several localization datasets (data filtered, lateral drift corrected, DAISY axial correction not applied) are available on Github as test samples for the DAISY correction code: https://github.com/ClementCabriel/DAISYcorrection. The authors also uploaded one clathrin-AF647 dataset obtained with DAISY (all corrections performed, data not filtered) on the

Shareloc platform: https://shareloc.xyz/#/view?u=z2Dig7bFraDdSHkXwg7Zhv. The authors will keep uploading datasets, both on Github and Shareloc. Other data are available from the corresponding authors upon reasonable request.

## Code availability

The localization and the lateral drift correction may be performed with any localization software. The DAISY correction code is available on Github at this address: https://github.com/ClementCabriel/DAISYcorrection.

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

## Acknowledgements

We thank Ultivue for consumable gifts and Abbelight for software and buffers gifts. We acknowledge the contribution of the Centre de Photonique BioMédicale to cell culture and labeling. We also acknowledge the help of Marion Bardou with cell culture. We thank Rym Boudjemaa for her contribution to the bacteria labeling project. Finally, we thank Caroline Schou and Yann Kerguttuil for their help regarding software analysis. This work was supported by the AXA research fund, the ANR (LABEX WIFI, ANR-10-LABX-24), the DIM CNANO Île-de-France, the IRS Bioprobe, the Mission interdisciplinarité of the CNRS, and LaserLab-Europe EUH2020654148. P.J. acknowledges a master funding from GDR ImaBio, and PhD funding from IDEX Paris Saclay (ANR-11-IDEX-0003-02).

## Author contributions

C.C., N.B., P.J., G.D., E.F. and S.L.F. conceived the project. C.C. designed the optical setup and performed the acquisitions. C.C. and N.B. carried out simulations and data analysis. P.J. and C.C. performed the CRLB calculations. N.B. developed the dSTORM buffer. N.B., C.C. and P.J. optimized the immunofluorescence protocol. P.J. and C.C. prepared the COS-7 cells samples, C.L. prepared the neuron samples, A.B., M.-A. B.-D. and B.V. prepared the bacteria samples. All authors contributed to writing the manuscript.

## Additional information

**Competing interests:** N.B., E.F., and S.L.F. are shareholders in Abbelight. The remaining authors declare no competing interests.

