## [Peer Review File · Nature Communications]

Reviewers' comments:

Reviewer #2 (Remarks to the Author):

Review of *Nature Communications* manuscript NCOMMS-18-35064-T

“Combining 3D single molecule localization strategies for reproducible bioimaging”

Cabriel et al. present Dual-view Astigmatic Imaging with SAF Yield (DAISY) as a 3D super-resolution technique that provides nearly constant 3D resolution over a 1 μ m range. DAISY splits the captured photons from each fluorophore into two channels: 1) a standard epifluorescence channel for 2D localization and measurement of sub-critical and super-critical photons and 2) an astigmatic channel for axial localization and measurement of sub-critical photons only. In this way, DAISY is able to extend DONALD's 3D measurement capabilities beyond 500 nm from the coverslip by using an astigmatic PSF. DAISY measures axial position with respect to the coverslip instead of the imaging system's focal plane, thereby reducing biases that stem from sample tilt, stage axial drift, and chromatic aberration. The authors characterize the precision of DAISY using fluorescent beads, showing 8 nm precision in xy and 12 nm precision along z. They demonstrate the correction of axial drift, chromatic aberrations, sample tilt, and astigmatism-induced image deformation within a 500 nm-1 μ m depth range using immunostained microtubules in COS-7 cells. Using sequential acquisitions taken at different z positions and cross-correlating them together, DAISY is shown to image *E. coli*, adducin-spectrin networks within neurons, and microtubules and clathrin spheres in COS-7 cells at various depths (200 nm up to 1500 nm).

In my opinion, the authors have an interesting solution for removing experimental biases in super-resolution imaging, that is, leveraging an absolute 3D coordinate system for improved accuracy. In this revised manuscript, the authors have addressed nearly all of the concerns raised in the previous review, and only minor issues remain. Detailed recommendations for improvement are below.

1. I thank the authors for providing a detailed description of their methodology for calculating the Cramér-Rao Lower Bound of the localization precision of SAF, astigmatic imaging, and DAISY. To place these calculations (which are used to justify the methodology for how to merge SAF and astigmatic channels) in context, several additional details or calculations should be provided:
 - a. The acronym CRLB should be defined on first use (p. 2, line 88).
 - b. One or two sentences should appear in the main text describing why the experimental measurements do not achieve the CRLB limit of localization precision. The CRLB is used to compare the performance of various SMLM methods, and achieving the CRLB is becoming standard for demonstrating new methods in SMLM (see refs. 7, 8, 19, 21, 28). Therefore, scientists looking to adopt DAISY will use the CRLB in this paper to compare this method to others, and should be aware of the potential challenges in achieving the CRLB presented in this paper. Possible reasons include optical aberrations and the use of centroiding and least-squares fitting instead of maximum-likelihood estimation, and possibly channel registration error mentioned below.
 - c. To further place DAISY in context with other 3D SMLM methods, the authors should add an SI figure that compares the CRLB for DAISY for N photons against that of SAF alone and astigmatism alone for N photons, not N/2 photons as plotted in Fig. 1b. That is, scientists looking to adopt DAISY will most likely be utilizing a single-channel imaging system, so the available number of photons is twice that of those available to the authors in this study,

who are using a two-channel imaging system, when using SAF or astigmatism independently. This additional data will clarify the tradeoff in localization precision necessary for implementing DAISY for removing experimental artifacts.

2. The authors perhaps misunderstood my previous concern regarding channel registration, which was mainly about accuracy and not precision. The authors did not include any discussion on registering the xyz measurements between the SAF and astigmatism channels (how to obtain x, y, and z in equations 1, 2, and 3 in the “position merging” methods section). In particular, this process could be adversely affected by field-dependent aberrations, which could be a source of uncorrected error in this study, as studied in ref. 23 (Diezmann et al.).
 - a. Please describe how the coordinate systems for x^{UAF} , x^{EPI} , y^{UAF} , and y^{EPI} were aligned with one another, since they begin as x and y positions measured on two separate regions of a camera. Quantify discrepancies in alignment between the UAF and EPI channels across the field of view for one of the datasets presented in the paper. Are registration errors constant across the field of view, or are field-dependent aberrations observed?
 - b. The axial cross-correlation algorithm used by the authors implicitly assumes that no field aberrations are present. Please add an additional SI figure discussing the axial cross-correlation performance for one of the datasets presented in the paper. What is the mean registration error in z positions between the SAF and astigmatism channels once their axial localizations are cross-correlated? How does this value change for different samples and different slices for extended-depth imaging? Are the authors able to detect field-dependent aberrations in the axial measurements between the two channels, as in ref. 23?
3. On page 5, line 203, the authors explicitly claim that “an arbitrary number of slices can be recorded and merged together” and implicitly claim that this task can be performed accurately for any sample thickness. To me, this statement is too strong without supporting evidence that shows that errors do not propagate along the sequential process of aligning neighboring stacks one-by-one (see my comment 2b above). I believe the discussion here should be amended to include the possibility of increasing error as more slices are added and reflect the need for absolute nanoscale calibration “rulers” on the scale of several micrometers to verify 3D depth accuracy.
4. Please correct the following typographical error:
 - a. Page 4, line 150: percents → percent

Reviewer #3 (Remarks to the Author):

In the revised manuscript, the authors addressed criticisms from this reviewer, which includes two key points, mainly by explaining in the rebuttal letter.

The first point is the impact of DAISY, a new microscopic method reported here. This reviewer found this concern common to all three reviewers. From the previous version of manuscript, this reviewer has recognized the performance of the combination of DONALD and the astigmatism, and the reviewer acknowledged the clarification of the benefit of this technique in the revised manuscript by adding Supplementary Fig. 4. However, in the rebuttal comment, the authors clearly stated that "The main message of this article is that the SAF axial detection challenges the paradigm of the widespread astigmatism technique" and "this paper is about revealing to the community the shortcomings of the widespread astigmatism golden standard using SAF and simultaneously endow astigmatism with pivotal new functionalities." These statements made it clear that the new method is developed aiming to improve already existing microscopic techniques by combining two methods. It is definitely worth publishing, but the topic is deep in developing microscopic technology and hence publishing in a specialized journal might be more appropriate.

The second point is about the evaluation of precision, which was also common point of interest for all three reviewers. This reviewer together with reviewer 1 was claiming that, the evaluation of the precision with real biological samples is more important than the estimation with fluorescent beads. The authors appealed by stating that "using beads (gold or fluorescent) to evaluate the axial localization precision is very common in the community provided they emit a number of photons similar to usual fluorophores" and giving some papers as examples. However, this appeal is not very accurate. Shtengel et al. (the first example paper) described the precision based on FWHM of real samples labeled with fluorescent proteins. Izeddin et al. (the second example paper) described precisions for both fluorescent proteins and dyes. In the first line of abstract, there is a statement as follows: "a slowly varying localization precision over a 1 μm range with precisions down to 15 nm". This is the only statement related to precision of DAISY in the abstract, and the value is with beads. Although the precision determined by biological samples were lower, this point is rather unclear throughout the manuscript. In the case of E. coli bacteria membrane, the standard deviation (σ) at a deep area ($z \approx 1 \mu\text{m}$) was 42-45 nm, and in addition, 68% confidence region is in theory between $-\sigma$ to $+\sigma$, namely 2σ (=82-90 nm), although the authors did not concern this point. The statement in the abstract has a risk misleading in terms of the achievable precision of DAISY especially in biological sample. Furthermore, for the experiment evaluating the precision with clathrin (Fig. 4gh) densities of signal points were quite low, which gave the impression that it is difficult to analyze their distribution precisely. Convincing data indicating the process determining FWHM, most probably based on axial and lateral histograms with sufficient density of signal points, should be included.

The authors wish to thank the reviewers for the helpful comments that help to strengthen the manuscript. You will find enclosed a point by point answer to those remarks. We also included the modifications added in the main text or supplementary, as well as the added figures when needed.

Reviewer #2

Cabriel et al. present Dual-view Astigmatic Imaging with SAF Yield (DAISY) as a 3D super-resolution technique that provides nearly constant 3D resolution over a 1 μ m range. DAISY splits the captured photons from each fluorophore into two channels: 1) a standard epifluorescence channel for 2D localization and measurement of sub-critical and super-critical photons and 2) an astigmatic channel for axial localization and measurement of sub-critical photons only. In this way, DAISY is able to extend DONALD's 3D measurement capabilities beyond 500 nm from the coverslip by using an astigmatic PSF. DAISY measures axial position with respect to the coverslip instead of the imaging system's focal plane, thereby reducing biases that stem from sample tilt, stage axial drift, and chromatic aberration. The authors characterize the precision of DAISY using fluorescent beads, showing 8 nm precision in xy and 12 nm precision along z. They demonstrate the correction of axial drift, chromatic aberrations, sample tilt, and astigmatism-induced image deformation within a 500 nm-1 μ m depth range using immunostained microtubules in COS-7 cells. Using sequential acquisitions taken at different z positions and crosscorrelating them together, DAISY is shown to image *E. coli*, adducin-spectrin networks within neurons, and microtubules and clathrin spheres in COS-7 cells at various depths (200 nm up to 1500 nm).

In my opinion, the authors have an interesting solution for removing experimental biases in superresolution imaging, that is, leveraging an absolute 3D coordinate system for improved accuracy. In this revised manuscript, the authors have addressed nearly all of the concerns raised in the previous review, and only minor issues remain. Detailed recommendations for improvement are below.

We thank again the reviewer for his/her constructive comments which helped to significantly improve the manuscript. We agree to all the following recommendations and give below detailed answers.

1-I thank the authors for providing a detailed description of their methodology for calculating the Cramér-Rao Lower Bound of the localization precision of SAF, astigmatic imaging, and DAISY. To place these calculations (which are used to justify the methodology for how to merge SAF and astigmatic channels) in context, several additional details or calculations should be provided:

a. The acronym CRLB should be defined on first use (p. 2, line 88).

We corrected this in the new version of the manuscript.

b. One or two sentences should appear in the main text describing why the experimental measurements do not achieve the CRLB limit of localization precision. The CRLB is used to compare the performance of various SMLM methods, and achieving the CRLB is becoming standard for demonstrating new methods in SMLM (see refs. 7, 8, 19, 21, 28). Therefore, scientists looking to adopt DAISY will use the CRLB in this paper to compare this method to others, and should be aware of the potential challenges in achieving the CRLB presented in this paper. Possible reasons include optical aberrations and the use of centroiding and leastsquares fitting instead of maximum-likelihood estimation, and possibly channel registration error mentioned below.

We added a few sentences (lines 94-97) to explain the fact that the experimental precisions do not reach the CRLB:

"It is worth noticing that the experimental precisions are slightly superior to the CRLB, which represent a theoretical ideal. This discrepancy is most likely due to optical aberrations, which are not taken into account by the CRLB, and to the use of centroid detection (see Methods), which is not expected to reach the lower limit."

c. To further place DAISY in context with other 3D SMLM methods, the authors should add an SI figure that compares the CRLB for DAISY for N photons against that of SAF alone and astigmatism alone for N photons, not N/2 photons as plotted in Fig. 1b. That is, scientists looking to adopt DAISY will most likely be utilizing a

single-channel imaging system, so the available number of photons is twice that of those available to the authors in this study, who are using a two-channel imaging system, when using SAF or astigmatism independently. This additional data will clarify the tradeoff in localization precision necessary for implementing DAISY for removing experimental artifacts.

Following Reviewer #2 suggestion on the need to compare DAISY with standard astigmatism, we added the **Supplementary Fig. 3**, which presents the CRLB values obtained for DAISY on the one hand, and for standard astigmatism (i.e. with a 300-nm spacing between the two focal lines, and with N photons contributing to the astigmatic detection instead of $N/2$) on the other hand. Although the theoretical precisions near $z = 0$ are satisfactory (5 nm lateral precision, 16 nm axial precision) with standard astigmatism, they rapidly deteriorate away from the focus plane (31 nm axial precision after 250 nm defocus). In comparison, DAISY offers minimal loss of performance at the focus, and a strongly enhanced precision in volume.

Supplementary Figure 3: Comparison of the lateral and axial CRLB for DONALD, standard astigmatism and DAISY as a function of the depth. The number of photons is 2750 for the UAF PSFs and 2750-5100 (depending on the axial position) for the EPI PSFs (similar to AF647), and the standard astigmatism corresponds to a 300-nm spacing between the two focal lines. The focus position is assumed to be 400 nm above the coverslip (typical experimental value), which is represented by the red dotted line. The solid and dashed lines stand for the axial and lateral precisions respectively. See **Supplementary Note 2** for an explanation of the CRLB calculations.

2. 2. The authors perhaps misunderstood my previous concern regarding channel registration, which was mainly about accuracy and not precision. The authors did not include any discussion on registering the xyz measurements between the SAF and astigmatism channels (how to obtain x , y , and z in equations 1, 2, and 3 in the “position merging” methods section). In particular, this process could be adversely affected by field-dependent aberrations, which could be a source of uncorrected error in this study, as studied in ref. 23 (Diezmann et al.).

a. Please describe how the coordinate systems for X_{UAF} , X_{EPI} , Y_{UAF} , and Y_{EPI} were aligned with one another, since they begin as x and y positions measured on two separate regions of a camera. Quantify discrepancies in alignment between the UAF and EPI channels across the field of view for one of the datasets presented in the paper. Are registration errors constant across the field of view, or are field-dependent aberrations observed?

Following the suggestion of Reviewer#2, we studied the lateral registration error more exhaustively. Especially, we added the **Supplementary Fig. 7**, in which we evaluate the discrepancies between the UAF and EPI lateral positions after the correction. We first performed this measurement on dark red fluorescent beads, which provide the most unambiguous results to our opinion: the mean discrepancy was found to be 7 nm (**Supplementary Fig. 7 a1-a2**). As a comparison, we performed the same measurement without the cylindrical lens, i.e. in a DONALD configuration, which yielded discrepancies around 4 nm (**Supplementary Fig. 7 a3**). We

analyze these results as a proof that our correction algorithm provides a correction precision (4 nm) significantly lower than the localization precision (10 nm at best), but we also hypothesize that our lateral localization could exhibit a slight PSF shape-dependent bias that would induce higher discrepancies for aberrated PSFs—the centroid detection is known to be biased by the coma aberration for instance. Another explanation would be a residual effect of the localization precision (which is significantly better with unastigmatic PSFs). In any case, one should keep in mind that potential field distortions are bound to be also present in standard (i.e. single channel) astigmatic imaging, in which case they can be neither detected nor corrected because of the lack of reference channel. Thus, our dual-view system allows to mitigate this bias, if not correct it fully. Note that the **Supplementary Fig. 7 a1** does not seem to show any regular pattern (i.e. the residual registration error does not seem to be field-dependent), which we interpret as a proof that the correction algorithm does not leave uncorrected translation, rotation or magnification difference between the UAF and EPI channels.

To further evidence the performance of our correction in terms of lateral registration error, we also measured the lateral discrepancies on a biological sample: we used the microtubules regions of interest presented in **Fig. 2e** to plot the histograms on both channels after correction. This gave values below 6 nm, which is consistent with the values obtained on fluorescent beads.

A reference to this supplementary material has been added on line 181 of the main text.

Supplementary Figure 7: Measurement of the residual lateral registration error after the correction. **(a)** Measurement on 40-nm diameter dark red fluorescent beads deposited at the coverslip (three acquisitions on different fields were stacked): **(a1)** Map of the residual lateral error, **(a2)** Residual lateral error histogram obtained with DAISY. As a comparison, the same measurement is provided for a DONALD acquisition (i.e. the same dual-view setup without the cylindrical lens) in **(a3)**. In both cases, the axial positions were averaged over 500 frames to mitigate the influence of the localization precision. The residual discrepancies are slightly superior for DAISY (7 nm) than for DONALD (4nm). We attribute this difference to either a PSF shape-dependent lateral bias (which would be larger for aberrated PSFs), or a residual influence of the localization uncertainty. **(b–c)** Measurements performed on the two microtubules regions of interest presented in **Fig. 2e**: **(b1–c1)** Superimposed 2D maps (red: EPI path, cyan: UAF path) before running the correction algorithm. The images display large discrepancies (around 500 nm) due to the magnification difference between the x and y axes. **(b2–c2)** Superimposed 2D maps after running the correction, **(b3–c3)** EPI and UAF profiles along the axes displayed in **(b2–c2)** after correction. The residual discrepancy is below 6 nm. Scale bars: 5 μm **(a)**, 1 μm **(b)**.

b. The axial cross-correlation algorithm used by the authors implicitly assumes that no field aberrations are present. Please add an additional SI figure discussing the axial crosscorrelation performance for one of the datasets presented in the paper. What is the mean registration error in z positions between the SAF and astigmatism channels once their axial localizations are cross-correlated? How does this value change for different samples and different slices for extended-depth imaging? Are the authors able to detect field-dependent aberrations in the axial measurements between the two channels, as in ref. 23?

Similarly, we added the **Supplementary Fig. 6** to study the accuracy of the axial cross-correlation between the SAF and astigmatism positions. With beads deposited on a coverslip, we obtain a mean axial discrepancy below 1 nm after applying the correction on the astigmatic positions. With beads distributed over [0-500nm] imaging range, we obtained a mean value below 3 nm. These values are significantly below the localization precision, and indicate that the accuracy of the cross-correlation is not limiting.

Supplementary Figure 6: Measurement of the residual axial registration error after the correction. **(a)** Histogram of the residual axial discrepancies between $z^{\text{astigmatic}}$ and z^{SAF} obtained with 40-nm diameter dark red fluorescent beads deposited at the coverslip. **(b)** Histogram of the residual axial discrepancies between $z^{\text{astigmatic}}$ and z^{SAF} obtained with 40-nm diameter dark red fluorescent beads randomly distributed in the volume. In both cases, the axial positions were averaged over 500 frames to mitigate the influence of the localization precision.

The histograms exhibit a certain dispersion of the values around the mean value, however (the measured standard deviations are 24 nm and 29 nm respectively for beads on the coverslip and in the volume). This could be due to the fact that the beads have a diameter of 40 nm, hence they are not point sources and the spot shapes are not exactly PSFs—in this case, better results could be obtained using a sample of sparsely distributed docking sites for DNA-PAINT labels in the solution, although we are not able to produce such a sample. Another possible explanation is the influence of field-varying aberrations. This effect is not specific to our method, as aberrations are known to induce field-dependent biases, but these biases are difficult to measure in conventional PSF shaping imaging as they require the use of calibration samples—as a result, their effect is scarcely documented in the literature. Considering the low values of the mean axial discrepancy, field-dependent biases do not seem to hamper the axial cross-correlation; on the contrary, the use of the complementary SAF detection reduces the impact of these biases since the SAF is not PSF shape dependent. The Reviewer further asks whether it is possible to use the SAF as a reliable reference to detect these spatially varying biases. We agree that it may be possible to perform a spatially resolved correction of the astigmatism calibration lookup table using the SAF positions as references. It should be noted, however, that this would only provide a correction for positions below 300 nm, as the SAF information is not available beyond that. Moreover, this would only give an offset value, while the aberrations are expected to modify the slope of the calibration curve too, as studied in [Diezmann et al., *Optica* (2015)]. Consequently, we believe that the best way to obtain the correct calibration is to use a specific calibration sample and to express the results as a (x, y, z) -dependent value instead of a z -dependent, in the same way as what Diezmann et al. proposed. Although such a sample would undoubtedly bring an improvement, we are not able to produce it because of its fabrication complexity. More generally, as the main point of DAISY is the correction of the axial drift, the

chromatic aberration and the sample tilt, we did not focus our efforts on actively reducing field aberration induced biases.

The main text was modified accordingly lines 158-168 :

“To illustrate the accuracy of the axial correction of the astigmatism data using the SAF measurement, we performed measurements on 40-nm fluorescent beads, both at the coverslip and distributed in the volume (Supplementary Fig. 6). In both cases, the axial correction algorithm seems very accurate (1 nm average discrepancy at the coverslip, and 3 nm in the volume, which is well below the localization precision). As the dispersion of the values increases for beads in the volume, this can be attributed to either the decay of the SAF signal in the volume, which causes the SAF localization precision to become non-negligible, or the influence of the previously mentioned field-dependent aberrations, which induce biases in the astigmatic positions according to the position in the field. This effect is present in conventional single-view PSF shaping imaging too, but it is difficult to detect unless a specifically designed calibration sample is used. The dispersion due to field-dependent aberrations could be mitigated by using a spatially resolved PSF calibration, as in [23].”

3. On page 5, line 203, the authors explicitly claim that “an arbitrary number of slices can be recorded and merged together” and implicitly claim that this task can be performed accurately for any sample thickness. To me, this statement is too strong without supporting evidence that shows that errors do not propagate along the sequential process of aligning neighboring stacks one-by-one (see my comment 2b above). I believe the discussion here should be amended to include the possibility of increasing error as more slices are added and reflect the need for absolute nanoscale calibration “rulers” on the scale of several micrometers to verify 3D depth accuracy.

We agree that claiming that an arbitrary number of slices can be acquired may be overstated. Indeed, the photobleaching is likely to limit the total acquisition time. This could be mitigated by using DNA-PAINT labeling, but even then, the lateral and axial precisions are expected to deteriorate deep in the volume (after a few μm) due to sample-induced aberrations. Finally, the addition of registration errors may lead to biases, and the use of fiducial markers at known depths may be required to merge many slices. These limitations were added in the manuscript at lines 228-231.

“Several slices can be recorded and merged together to obtain an extended depth image—still, this is limited by photobleaching (although this can be mitigated by using (DNA-PAINT labeling), as well as aberrations inherent in depth imaging, which cause the axial and lateral precisions to deteriorate away from the coverslip”

4. Please correct the following typographical error: Page 4, line 150: percents → percent

The spelling mistake was corrected (line 172).

Reviewer #3

In the revised manuscript, the authors addressed criticisms from this reviewer, which includes two key points, mainly by explaining in the rebuttal letter.

The first point is the impact of DAISY, a new microscopic method reported here. This reviewer found this concern common to all three reviewers. From the previous version of manuscript, this reviewer has recognized the performance of the combination of DONALD and the astigmatism, and the reviewer acknowledged the clarification of the benefit of this technique in the revised manuscript by adding Supplementary Fig. 4. However, in the rebuttal comment, the authors clearly stated that “The main message of this article is that the SAF axial detection challenges the paradigm of the widespread astigmatism technique” and “this paper is about revealing to the community the shortcomings of the widespread astigmatism golden standard using SAF and simultaneously endow astigmatism with pivotal new functionalities.” These statements made it clear that the new method is developed aiming to improve already existing microscopic techniques by combining two methods. It is definitely worth publishing, but the topic is deep in developing microscopic technology and hence publishing in a specialized journal might be more appropriate.

Reviewer #3 is not convinced that the work presented in the manuscript it is suited for a publication in *Nature Communications* as he/she questions its impact. The main reason for this assessment is that the technique presented appears somewhat specific to him/her or that the message is not broad range enough: he/she writes “the topic is deep in developing microscopic technology and hence publishing in a specialized journal might be more appropriate.” The authors want to point out that 3D super-localization microscopy is spreading rapidly and the large number of applications to biological or material physics studies contribute to increasing the need for new developments. Super-resolution not anymore only concerns the optics and physics communities, but also a large interdisciplinary community of users. Still, the literature seems very scarce to us when it comes to highlighting the shortcomings of widespread techniques. Notable exceptions include [Diezmann *et al.*, *Optica* (2015)], [Backlund *et al.*, *Nature Photonics* (2016)] and [Aristov *et al.*, *Nature Communications* (2018)], but articles presenting both an exhaustive study of the limitations of existing methods and the solutions to overcome these remain quite rare. Our approach in this work is mostly to evidence the bottlenecks common to the vast majority of 3D nanoscopy methods and to propose an optically simple solution to overcome them—which is crucial to obtain precise, reliable and reproducible results. Indeed, the experimental setup is more straightforward than in most 3D methods, and the correction code is thoroughly detailed in the manuscript and released as an open access Python script to make it available to the community alongside with several data sets to test it (<https://github.com/ClementCabriel/DAISYcorrection>). Consequently, it seems highly desirable to bring both this analysis of the shortcomings of currently used techniques and this solution to the knowledge of a broad readership belonging to several scientific communities. Specialized journals would definitely reduce the impact of this broad range message compared with multidisciplinary high impact journal like *Nature Communications*.

The second point is about the evaluation of precision, which was also common point of interest for all three reviewers. This reviewer together with reviewer 1 was claiming that, the evaluation of the precision with real biological samples is more important than the estimation with fluorescent beads. The authors appealed by stating that “using beads (gold or fluorescent) to evaluate the axial localization precision is very common in the community provided they emit a number of photons similar to usual fluorophores” and giving some papers as examples. However, this appeal is not very accurate. Shtengel *et al.* (the first example paper) described the precision based on FWHM of real samples labeled with fluorescent proteins. Izeddin *et al.* (the second example paper) described precisions for both fluorescent proteins and dyes. In the first line of abstract, there is a statement as follows: “a slowly varying localization precision over a 1 μm range with precisions down to 15nm”. This is the only statement related to precision of DAISY in the abstract, and the value is with beads. Although the precision determined by biological samples were lower, this point is rather unclear throughout the manuscript. In the case of *E. coli* bacteria membrane, the standard deviation (σ) at a deep area ($z \approx 1 \mu\text{m}$) was 42-45 nm, and in addition, 68% confidence region is in theory between $-\sigma$ to $+\sigma$,

namely 2σ (=82-90 nm), although the authors did not concern this point. The statement in the abstract has a risk misleading in terms of the achievable precision of DAISY especially in biological sample. Furthermore, for the experiment evaluating the precision with clathrin (Fig. 4gh) densities of signal points were quite low, which gave the impression that it is difficult to analyze their distribution precisely. Convincing data indicating the process determining FWHM, most probably based on axial and lateral histograms with sufficient density of signal points, should be included.

As for the estimation of the localization precision, the Reviewer expresses concerns about our appeal, stating: "Shtengel et al. (the first example paper) described the precision based on FWHM of real samples labeled with fluorescent proteins". The Reviewer further stresses the fact that, in the second article (Izeddin et al.), the authors mention that the localization precision was evaluated first from fluorescent beads and then from biological samples. In fact, both articles follow the same approach, commonly encountered for the development of new instrument: after measuring the localization precision performances on sparse, isolated sources emitting controlled numbers of photons (isolated fluorescent proteins in Fig. 2A in Shtengel et al. (represented below), gold nanobeads in Fig. 2B and Fig. S1A in Shtengel et al., fluorescent nanobeads in Fig. 3 in Izeddin et al.), they study biological samples and find comparable levels of precisions (Fig. 3-4-5 in Shtengel et al., Fig. 4 in Izeddin et al.).

Figure 2 from Shtengel et al.

This image is not covered by the article CC BY license. Image credit to PNAS. All rights reserved, used with permission

Fig. 2. X, Y, Z resolution of iPALM and its dependence on source brightness, illustrating iPALM's sub-20-nm 3D resolution with endogenous FP labels. (A) A histogram of experimentally determined positions from repeatedly sampling (25,000 frames) a source where $\approx 1,200$ photons are detected per frame from $\approx 1,500$ photons emitted into a 4π solid angle. (B) Axial (solid red circles) and lateral (solid blue squares) resolution of iPALM determined from FWHM of localization of Au beads of different brightness. Note that the positional FWHM number is 2.4 times larger than σ the variance that is also used to characterize resolution. Axial (empty red circles) and lateral (empty blue squares) resolution of the defocusing method determined from FWHM of localized position of Au beads of different brightness. Large ovals indicate approximately the published results for axial (red) and lateral (blue) resolutions of 3D STORM (2) and BP PALM (3). The typical photon output of fluorescent protein tags and synthetic fluorophores are depicted as pink and green gradients. Also shown (horizontal dashed lines) are additional uncertainties resulting from the displacement between the target protein and the fluorescent probes for different imaging methods.

We thus adopted the very same approach in this manuscript: we evaluate the performances first on fluorescent beads (Fig. 1c), and then on biological samples (E. coli bacteria in Fig. 3c and Supplementary Fig. 8, clathrin spheres in Fig. 4g-h and Supplementary Fig. 9, and finally microtubules in Supplementary Fig. 2).

The Reviewer also states that the localization precisions measured on biological samples are higher than those obtained on fluorescent beads. While the standard deviations of the localization distributions that we measure are definitely slightly narrower on fluorescent beads, it is most likely due to the size and the geometry of the label used (antibody labeling constructions typically exhibit sizes around 10-15 nm, while biotin-

streptavidin constructions are expected to measure roughly 10 nm) as well as to the fact that the immunolabeled proteins might not be punctual. More generally, we believe that measuring localization precisions using fluorescent beads is more reliable and relevant than using biological samples as the results do not rely on hypotheses made on the target size or on the geometry of the labels, which, incidentally, are not known precisely. We have added these explanations to the Methods (lines 487-491):

“Using fluorescent beads seems to be a more reliable method to measure the localization precisions than with biological samples—unlike the fluorescent beads, the use of biological samples require many assumptions on the size and geometry of the labeled target, the label (which is typically around 10-15 nm in the case of immunolabeling), the fluorophore itself, as well as the motion freedom of the label.”

We agree with the Reviewer that it is also important, as a second step, to evidence the precision in biological samples, and additional analysis of the data have been inserted. We added in **Fig. 4g-h** the axial histograms plotted on the clathrin samples. For a more thorough data analysis, we also added in **Supplementary Fig. 9** the lateral histograms as well which help to appreciate the resolution.

Figure 4g-h:

Supplementary Figure 9: Lateral and axial histograms plotted on the clathrin spheres presented in **Fig. 4g-h**. The histograms are plotted along lines at the center of the spheres to highlight their hollowness. Scale bars: 250 nm.

Finally, we also added the **Supplementary Fig. 2**, which presents lateral and axial profiles plotted on microtubules labeled with AF647. From those histograms, we measured the distance between the peaks. Using the method published in [Bourg et al., Nature Photonics (2015): Supplementary Fig. 7], we deduce from those measurements that the 3D localization precision for that experiment is around 14-16 nm, which is consistent with the values obtained on fluorescent beads, E. coli bacteria and clathrin spheres.

The main manuscript has been accordingly modified (lines 89-90):

“Such precision is sufficient to resolve the hollowness of immunolabeled microtubules, as displayed in Supplementary Fig. 2.”

Supplementary Figure 2: Visualization of the hollowness of microtubules. **(a)** 2D super-localized image of COS-7 cells with α -tubulin labelled with AF647. The acquisition is the same as that presented in **Fig. 2e**. The lateral (blue) and axial (red) histograms are then plotted in the green boxed region **(b)** and in the yellow boxed region **(c)**. Both the experimental data and the fitted profile with a double Gaussian function are displayed, as well as the distance between the two Gaussian peaks. The hollowness is clearly visible, and the distance between the peaks corresponds to a localization precision around 14-16 nm (see Ref. [1], **Supplementary Fig. 7**). Scale bar: 5 μ m.

Reviewer #3 later states that providing the $\pm 2\sigma$ (i.e. 95% confidence) intervals would result in wide range values. Although this is true, the convention in the domain is to provide localization precision values as the standard deviation of the histograms. A few publications also give Full Widths at Half Maximum (FWHM) as the precision standard, but to our knowledge, no article mentions the $\pm 2\sigma$. Providing such intervals would hence be misleading or confusing to most researchers used to reading super-localization articles. Nevertheless, in order to make our precision measurements clearer, we changed the manuscript to provide all the values as the standard deviation of the histogram: we thus modified the analysis of the clathrin spheres (lines 244-246), which was displaying FWHM rather than the standard deviations. In addition, at the request of the Reviewer, we added in **Fig. 4g-h** and **Supplementary Fig. 9** the lateral and axial histograms, as we did for the E. coli bacteria in **Fig. 3c**. Please note that the numbers of localizations seem large enough to plot the distribution of the localizations. Also, please note that the histogram standard deviations at 200 nm depth ($\sigma_{xy} = 16$ nm and $\sigma_z = 17$ nm) seem to be in good agreement with what can be expected taking into account the fact that, as mentioned previously, the size of the antibody constructions are around 10-15 nm.

Finally, we would like to emphasize again the fact that the localization precision per se is not as critical a point as the rest of the limitations inherent to 3D super-localization methods presented in our manuscript. Indeed, very precise methods are likely not to be used at their full potential if the setup is subject to drifts or chromatic aberrations, or if the label used to stain the sample is large. In these frequent cases, the results are bound to be limited simply by the defects of the setup or of the sample. The main point raised by our manuscript is thus to propose a solution to deal with non-idealities of the setup rather than achieving an ultimate localization precision, which would be wasted without these necessary corrections.

Reviewers' Comments:

Reviewer #2:

Remarks to the Author:

Review of *Nature Communications* manuscript NCOMMS-18-35064A

“Combining 3D single molecule localization strategies for reproducible bioimaging”

Cabriel et al. present Dual-view Astigmatic Imaging with SAF Yield (DAISY) as a 3D super-resolution technique that provides nearly constant 3D resolution over a ~600nm range. DAISY splits the captured photons from each fluorophore into two channels: 1) a standard epifluorescence channel for 2D localization and measurement of sub-critical and super-critical photons and 2) an astigmatic channel for axial localization and measurement of sub-critical photons only. In this way, DAISY is able to extend DONALD's 3D measurement capabilities beyond 500 nm from the coverslip by using an astigmatic PSF. DAISY measures axial position with respect to the coverslip instead of the imaging system's focal plane, thereby reducing biases that stem from sample tilt, stage axial drift, and chromatic aberration. The authors characterize the precision of DAISY using fluorescent beads, showing 8 nm precision in xy and 12 nm precision along z. They demonstrate the correction of axial drift, chromatic aberrations, sample tilt, and astigmatism-induced image deformation within a 500 nm-1 μ m depth range using immunostained microtubules in COS-7 cells. Using sequential acquisitions taken at different z positions and cross-correlating them together, DAISY is shown to image E. coli, adducin-spectrin networks within neurons, and microtubules and clathrin spheres in COS-7 cells at various depths (200 nm up to 1500 nm).

In this revised manuscript, the authors have addressed all of the concerns raised in the previous review, and only minor issues remain. I find DAISY to be an elegant way of correcting common experimental biases that cause artifacts in super-resolution microscopy, and once corrected, I believe the manuscript is suitable for publication in *Nature Communications*. Detailed recommendations for improvement are below.

- 1) The super-resolution image in Supplementary Figure 2a seems to depict the microtubule structures differently from the data in the cross-sectional histograms in Supplementary Figures 2bc. That is, the image does not seem to show hollow microtubules, but instead shows a faint “shadow” of a microtubule displaced above-right with respect to the main microtubule in both the yellow and green boxes. Please provide larger, zoomed insets of the yellow and green regions in panel (a) that clearly agree with the histograms in panels (b)-(c).
- 2) Line 85: Please correct the spelling of “depending.”
- 3) Lines 88 and 95: The authors' usage of “inferior” and “superior” does not seem to be consistent with standard English. I believe the authors intend to write: “they both remain better (less) than 20 nm” in line 88 and “precisions are slightly worse (larger) than the CRLB” in line 95. Please correct this terminology.

Reviewer #3:

Remarks to the Author:

In the revised manuscript the authors addressed all criticisms from this reviewer.

Once again, we thank the Reviewers for their feedback. Please find below our response to the constructive comments made by the Reviewer 2.

1) The super-resolution image in Supplementary Figure 2a seems to depict the microtubule structures differently from the data in the cross-sectional histograms in Supplementary Figures 2bc. That is, the image does not seem to show hollow microtubules, but instead shows a faint “shadow” of a microtubule displaced above-right with respect to the main microtubule in both the yellow and green boxes. Please provide larger, zoomed insets of the yellow and green regions in panel (a) that clearly agree with the histograms in panels (b)-(c).

The Reviewer rightly stresses this “shadow” effect, which we did not notice in the previous version of the manuscript. By studying the drift correction results data more thoroughly, we noticed that it came from a problem in the lateral drift correction at the very beginning of the acquisition. We thus removed the frames that were not adequately corrected from the localization list. The Supplementary Figure 2 was updated with the new localization list, and a zoomed inset was added (Supplementary Figure 2b) to show more clearly the hollowness directly on the 2D localization image.

The problem was present only in the Supplementary Figure 2a. The histograms (Supplementary Figure 2c-d) and the images presented in Figure 2i were generated from the right data set; as a consequence, we did not update them.

It is worth noticing that, while this faint “shadow” effect was visible on the image, the hollowness observed on the histograms comes from the sample itself, as the “shadow” image is too dim to cause a degradation of the data. To illustrate this, we represent in the figure below two histograms plotted on the same microtubule: one generated with the correct localization list (cyan) and one obtained from the wrong (red). Note that the shadow is visible on the red histogram in the dotted boxed area. Still, this is not sufficient to significantly degrade the visualization of the hollowness, which appears on both data sets.

2) Line 85: Please correct the spelling of “depending.”

The spelling mistake was corrected in the new version of the manuscript.

3) Lines 88 and 95: The authors’ usage of “inferior” and “superior” does not seem to be consistent with standard English. I believe the authors intend to write: “they both remain better (less) than 20 nm” in line 88 and “precisions are slightly worse (larger) than the CRLB” in line 95. Please correct this terminology.

The sentences were corrected in accordance with the comment of the Reviewer in the new version of the manuscript.